The pivotal role of software defined networks to safeguard against cyber attacks: a comprehensive review

Aljughaiman Ahmed
Almarri Seetah 224108483@student.kfu.edu.sa
Department of Computer Networks and Communications, College of Computer Sciences and Information Technology, King Faisal University , Al-Ahsa , Saudi Arabia
Givargis Tony
Electronic publication date: 2025 Apr 7
Publication date: 2025
Volume: 11
Electronic Location ID: e2814
Received 2024 Sep 23; Accepted 2025 Mar 19
Copyright: © 2025 Aljughaiman and Almarri
Copyright year: 2025
Copyright holder: Aljughaiman and Almarri
License: This is an open access article distributed under the terms of the Creative Commons Attribution License, which permits unrestricted use, distribution, reproduction and adaptation in any medium and for any purpose provided that it is properly attributed. For attribution, the original author(s), title, publication source (PeerJ Computer Science) and either DOI or URL of the article must be cited.
License URL: https://creativecommons.org/licenses/by/4.0/

Keywords: SDN, Cybersecurity, Cyber threats, Attacks, Countermeasures

Funding: Deanship of Scientific Research, Vice Presidency for Graduate Studies and Scientific Research, King Faisal University, Saudi Arabia KFU250905 This work was supported by the Deanship of Scientific Research, Vice Presidency for Graduate Studies and Scientific Research, King Faisal University, Saudi Arabia (GRANT No. KFU250905). The funders had no role in study design, data collection and analysis, decision to publish, or preparation of the manuscript.

==============================
Software defined networks (SDNs) offer novel approaches to managing networks by separating the control plane from the data plane to enable programmable control over network resources effectively and dynamically. This framework supports monitoring of traffic flow and detection of threats while also enabling easy adaptation of network configurations, which is critical in safeguarding against cyber threats. However, this separation also brings forth security risks that cyber attackers may exploit. In this examination, the basic concepts of SDN are explained, pointing out their benefits compared to conventional networks and exploring the security issues that are part of SDN architectures. Different types of threats that focus on SDN layers are categorized and how they impact network security while suggesting different ways to address them. Furthermore, the review highlights issues and suggests potential research paths to enhance SDN security measures and ensure their effectiveness against ever-changing cyber dangers.

Introduction

Over the last few years, SDN has become popular as a transformative network architecture, revolutionizing the telecommunications industry and playing a crucial role in initiatives such as 5G and the Internet of Things (IoT) (Jimenez et al., 2021). SDN is unlike conventional network architectures that lack the programmability and centralized control that SDN offers to improve the efficiency, security, and flexibility of the network. This is especially important in areas like cloud computing, 5G networks, and the IoT, where dynamic network control is critical for managing traffic complexity and security threats. The division of the control and data planes in SDN puts the control plane where it can be easily administered by the network administrators to help in better resource management, security enforcement, and threat detection and mitigation (Rahouti et al., 2022).

More flexibility, security, and agility are considered the most important features that SDN can provide to organizations by creating programmable networks. It addresses the challenges of traditional network infrastructures in handling unpredictable data traffic patterns by allowing dynamic network configuration, monitoring, and resource allocation. Cloud service providers (CSPs) unlike networks, acknowledge the innovative potential of SDN in building robust and scalable data center infrastructures (Jimenez et al., 2021). Through the separation of the data and control planes, SDNs introduce an Application Programming Interface (API), enabling the incorporation of programmable features into networks. This gives network service providers the ability to utilize manageable architectures (Farooq, Riaz & Alvi, 2023). Implementations of SDN like Microsoft’s public cloud service and the edge gateways from Nippon Telegraph and Telephone (NTTs), a global provider of business solutions and technology services company, along with the Google B4 network system have showcased the advantages of this network structure.

However, as SDN gains popularity, security concerns have also emerged. Authentication, authorization, access control, and protection against malicious applications are some of the security challenges faced by SDN networks. Ensuring the control layer’s security, where the SDN controller is located, becomes crucial to maintaining the confidentiality, integrity, and availability (CIA) of network operations. SDN represents a transformative approach to network architecture, offering programmability, flexibility, and centralized management (Farooq, Riaz & Alvi, 2023). It finds applications in various domains, improving network scalability and resource optimization. However, addressing security concerns is essential to safeguard SDN networks from potential threats. The purpose of this article is to give an overview of SDN, its benefits, and its impact on network configuration and performance. Furthermore, this review article focuses on the challenges facing the security of SDN and the necessity of taking security countermeasures to protect them from hacking and cyberattacks. The article provides researchers and practitioners with a foundational resource in the form of a unified perspective along with synthesizing fragmented knowledge to design resilient SDN architectures against evolving cyber threats.

There is a problem with the fact that current reviews of SDN security are designed to highlight certain threats or layers and therefore do not provide a systematic approach to the vulnerabilities of the entire SDN architecture. Most of the prior studies focus on specific planes of the SDN (e.g., control or data plane) or types of attacks (e.g., distributed denial-of-service (DDoS)) while neglecting the interconnectedness of SDN layers and interfaces. Also, most of the reviews do not use a very specific method of selecting literature, which may result in some form of bias or missing information. The purpose of this article is to conduct a systematic literature review (SLR) using the PRISMA framework to guide the process transparently and comprehensively when analyzing 20 high-quality studies within the timeframe of 2019–2024. This article makes three key contributions to address these gaps: Comprehensive taxonomy: A structured approach for threat identification along with its corresponding countermeasures through a comprehensive examination of all SDN planes (control, data, application) and interfaces (northbound, southbound, east/westbound).

Methodological rigor: A reproducible systematic literature review process that weakens bias and highlights neglected areas of research like standardizing the interface and the scalability issues in emerging technologies such as blockchain and AI.

Actionable future directions: Clear pathways to advance SDN security research where gaps are bridged.

This article structure is organized as follows: “Articles Selection For Literature Review” shows how the articles are selected and analyzed. “Background” clarifies the concept of SDN and its significant role in managing the network and highlights how SDN differs from traditional networks. “Related Study” presents a comparative study of the reviewed articles, identifies existing problems and methodologies used, and sheds light on the limitations. “Security Challenges and Countermeasures in SDN” describes attack classifications that target SDN layers and organize them based on various attack surfaces and countermeasures, which provide detailed strategies for addressing various types of attacks. In “Future Directions and Open Research Challenges”, potential future directions are presented. “Conclusions” the conclusion is presented.

Motivation

While there has been an increase in research on SDN, there is still a lack of reviews that not only classify security risks but also conduct a methodical examination of the available countermeasures. This study aims to address this gap by presenting an analysis of the security issues encountered in SDN and discussing possible remedies while pinpointing areas that require additional investigation. By giving this summary of the topic at hand, the article intends to help advance the creation of safer and stronger SDN structures.

Papers selection for literature review

Methodology

This study utilizes a systematic literature review (SLR) to organize and present information in an organized manner (Snyder, 2019). The SLR method is crucial for identifying gaps and constraints in existing research while also paving the way, for research directions. To illustrate the process of selecting articles, the PRISMA flowchart was employed with three key stages. Identification, screening, and inclusion.

In the phase of the identification process targeted studies released between 2019 and 2024 for examination while eliminating duplicate and unsuitable records based on the year and source type (like journals and conference articles). The screening stage then refined the selection by excluding records that were not relevant to the research topic and those, with length of content. The final step involved selecting articles that fulfilled all the review criteria.

Search string

The search query utilized to enhance the search outcome quality was (“Software Defined Network” OR “SDN”) AND (“Threats” OR “Challenges”) AND (“Solutions” OR “Countermeasures”). It involves operators such, as “AND” and “OR” connecting the keywords together which aids in expanding or refining the search query effectively.

Data sources

The search string was applied in two databases which are Google Scholar, and Saudi Digital Library.

Screening process

The articles are screened by using a search query while primarily looking at the titles to gauge their relevance to our subject matter. If a article’s relevance was uncertain based on its title, an extra screening measure was implemented by examining the abstract to better evaluate its appropriateness. The selection process for identifying and incorporating articles into this review is detailed in the PRISMA flowchart depicted in Fig. 1.

Figure 1 Selection of articles with PRISMA for a literature review.

Selection process

In this study, the PRISMA 2020 framework is used to help organize and manage data flow throughout the review process (Page et al., 2021). To start the research process, two main databases were used to find recent and relevant studies: Google Scholar, which gave 16,100 results, and the Saudi Digital Library, which returned 7,205 results. The following search query is used (“Software Defined Network” OR “SDN”) AND (“Threats” OR “Challenges”) AND (“Solutions” OR “Countermeasures”) to search. Also, the articles that were published between 2019 and 2024 were chosen.

A total of 19,509 records were removed throughout this process and prior to screening for a variety of reasons, including duplicate records or being flagged as ineligible. After a comprehensive review process, 3,796 records underwent detailed examination, resulting in the selection of 20 studies that aligned with the research goals. The remaining 3,776 records were excluded due to factors such as being irrelevant to the subject, outside the designated timeframe, written in a language other than English, or lacking accessibility.

Background

In SDN networks, the network provides centralized control and programmability by its feature of separating the data plane from the control plane while each network device in a traditional network uses its control plane to determine its decisions regarding the routing. In SDN, the data plane is distributed among network devices, but the control plane is abstracted and relocated to a centralized controller (See Fig. 2). Easier management, dynamic traffic engineering, and improved programmability are the main benefits of SDN architecture. In addition, there are a few security challenges that need to be addressed (Almadani, Beg & Mahmoud, 2021).

Figure 2 SDN and traditional network components.

Furthermore, the centralization of control is one of SDN’s features. When a single centralized controller manages the network infrastructure, it becomes vulnerable to attacks leading to access and traffic manipulation risks (Sahoo et al., 2019). To safeguard network integrity and prevent disruptions from a compromised controller (Sahoo et al., 2019), it is crucial to secure both the controller and its communication pathways effectively. Moreover, SDNs provide the flexibility for administrators to dynamically adjust network behavior by programming them with software applications (Iqbal et al., 2019). It is important to be careful because this functionality could bring about dangers like access to data and disruptions in service delivery. Ensuring that appropriate security measures are implemented is vital for verifying and approving program changes, detecting activities, and upholding security regulations throughout the SD network architecture.

The separation of control and data planes in SDN can also have implications for security measures. The controller requires a communication link to convey instructions to network devices by bridging the data plane and control plane. However, intruders might attempt to intercept or alter instructions within the data plane by focusing on this communication channel (Shaghaghi et al., 2020). Such attacks could result in data leakage tampering with traffic patterns enabling network access and other potential risks.

Addressing security threats and vulnerabilities in SDNs is crucial, as the programmability and centralized control they offer can enhance network visibility and control but also leave them susceptible to exploitation by actors. A multi-layered security approach should be implemented in order to reduce these threats which including intrusion detection systems (IDS), secure controller design, robust access controls, encrypted communication routes, and frequent security audits. SDN security-specific standards, best practices, and recommendations can be followed to greatly increase the resilience of SDN deployments and guard against potential threats. In the following subsections, SDN architecture are explained in much more detail.

SDN architecture and components

SDNs are a type of network design that enables management and control by separating the data processing and control functions of a network system. This portion is intended to detail the principles and elements of SDNs, like the SDN controller, control layer, application layer, data layer, and their interconnections. It also discusses the OpenFlow (OF) protocol (Wazirali, Ahmad & Alhiyari, 2021), frequently employed to facilitate communication between switches and controllers in SDN as illustrated by Fig. 3.

Figure 3 SDN architecture.

Application plane: At the top of the SDN architecture is the application plane, also known as the management plane. Controller’s developers or third parties produce a variety of business and security applications, and this layer serves as their execution platform. Implementing firewalls, access controls, load balancing, routing policies, intrusion prevention systems (IPS), IDS, and network virtualization are among the applications that this layer manages (Shaghaghi et al., 2020).

Control plane: The control plane plays the role of a middleman between the application layer and the data layer within the SDN framework by ensuring network traffic forwarding and overseeing network operations overall using an SDN controller as the central brain (Chai et al., 2019). This control plane includes the controller for managing network behavior and diverse network applications offering a variety of services to communicate with switches in the data plane to adjust their forwarding actions.

SDN controller: The SDN controller provides a logical view and control over the whole network, making it the key and most important part of the SDN architecture. It makes it easier to communicate with applications via the northbound interface and the data plane via the southbound interface. The controller communicates with switches in the data plane, issuing instructions to configure their forwarding behavior and exchanging data. Network programs that provide extra network services like load balancing, security, and traffic engineering are also hosted by the controller (Zhu et al., 2020).

There are three types of controllers which known as centralized single controller, flat distributed controller, and hierarchical distributed controller (as shown in Fig. 4). Flat and hierarchical subtypes are two further classifications for the distributed controller (Zhu et al., 2019). The different tasks are given to each controller in the SDN architecture. These controller types are elaborated in Table 1.

Data plane: Another name of data plane is the forwarding plane, takes on the vital responsibility of actually forwarding network traffic in the SDN architecture. It consists of routers or network switches that use the controller’s instructions to find the best path for packets or frames to be forwarded (Kaljic et al., 2019). The controller establishes predetermined rules or policies that the data plane uses to carry out the forwarding procedure.

Data planes can be classified into two categories: stateless and stateful (Zhang et al., 2021). Network components don’t hold onto network states in a stateless data plane. When they need to execute any new activities, they must ask the controller, as they carry out the decisions made in the control plane. If dynamic network behavior is required, the control plane helps assign tasks to the data plane. Data planes that are stateless can become stateful by help of this delegation, which grants the data plane the ability to store network states and perform actions (Zhang et al., 2021).

Forwarding decisions, updating and topology discovery are essential tasks within data plane. Two essential services are mostly needed for these tasks: link discovery service (LDS) and host tracking service (HTS) (Deb & Roy, 2022).

The link layer discovery protocol (LLDP) is used by LDS to collect switch and inter-switch link details. Through the openFlow discovery protocol (OFDP), link information between OF switches is retrieved in SDN networks that use OF. Alternatively, the broadcast domain discovery protocol (BDDP) is used to gather link information between OF switches and traditional switches (Deb & Roy, 2022). The LLDP protocol is the source of both OFDP and BDDP. On the other hand, The HTS keeps track of the network’s hosts and their respective locations. The exchange of packet-in and packet-out messages during the switch-controller handshake is the main function of these services. The intervals of these messages between entities are set by each controller. The controller can efficiently manage network resources and allow traffic rerouting based on real environmental requirements by using the updated topology information. This improves the quality of service (QoS) offered to the applications in the application plane and permits better path discrimination when necessary (Deb & Roy, 2022).

Northbound API: The Northbound API in SDN is the interface that located between the application layer and the SDN controller. By acting as a network abstraction interface, this API gives management systems and applications running at the top of the SDN a consistent way to communicate with the underlying network infrastructure. Furthermore, the Northbound API is not standardized, each controller specifies APIs that should be used. Right now, the most used API for applications is REST (Latif et al., 2020).

Southbound API: The Southbound API is the interface that located between the SDN controller and the data plane in a distributed data network. Routers can use this API to implement requests received via the northbound interfaces, determine network flows, and determine network topology. OF is the most widely standardized and used Southbound API and Open vSwitch Database (OVSDB) is another example (Latif et al., 2020).

OpenFlow protocol: A standardized communication protocol utilized between the switches in the data plane and the SDN controller is called OF. It enables the controller to communicate with the network’s forwarding elements. The OF protocol specifies a collection of commands and messages that allow the controller to configure forwarding rules, collect network traffic statistics, and receive event notifications from the switches. The network can be managed and controlled centrally using this protocol (Latif et al., 2020).

There is no security protection in OF. The OF switch and the OF controller can optionally establish secure communication connections using plain transmission control protocol (TCP) or transport layer security (TLS) for the primary connections, user datagram protocol (UDP), TCP, or datagram transport layer security (DTLS) for the secondary connections. TLS is advised by the open networking foundation (ONF) starting with version 1.2 (Yigit et al., 2019).

East/Westbound API: The east/westbound interface uses the Eastbound API to enable the connectivity of distributed controllers. One of the main features of SDN is centralized network control. To provide efficient coordination across dispersed controllers, Eastbound APIs are used for information import and export (Latif et al., 2020). On the other hand, Westbound APIs facilitate communication between the controllers and legacy network components like routers, guaranteeing a smooth transition between SDN and conventional networking components.

Figure 4 Types of controllers.

Table 1 Types of controllers.

	Centralized controller	Flat controller	Hierarchical controller	
Description	A single controller that manages all control plane operations across the network.	Throughout the network, several controllers are placed, and every controller is responsible for a certain number of network devices.	Many levels of controllers, each level responsible of a particular domain.	
Key features	Central point of control for the whole network and consistent decision making based on the overall status of the network.	Decentralized control and load balancing	Fault tolerance, high performance, better management, network resilience, throughput, latency and scalability.	
Limitations	Potential to a single point of failure and scalability problems	Lack of global view and coordination challenges	Difficult synchronization, increased complexity and increasing in the operational cost.	
Threats	An attacker may take over the entire network if it is compromised, which could result in service interruptions, data breaches, or unauthorized access.	It gets harder to implement consistent security policies across the network in the absence of centralized control.	Inter-layer attacks.	

The application plane, control plane, and data plane are the three main planes that make up the SDN architecture as shown in Fig. 3. The physical network components that make up the data path are contained in the data plane. SDN controller, which is part of the control plane, applies rules to data plane devices. The SDN architecture’s application plane is where these guidelines and policies are created. APIs that are well specified are used to establish communication across these planes. These interfaces are separated into East/Westbound APIs, Northbound APIs, and Southbound APIs in the case of distributed controllers. Southbound API is used for communication between the control and data planes. It transmits data plane device information to the control plane and pushes rules or instructions from the control plane to the data plane devices. Northbound API is used by the control plane and application plane to enable programmability in SDN. Eastbound API is used to establish inter-controller communication across SDN domains, while Westbound API handles legacy domain to SDN domain connection.

Advantages of SDN architecture

SDN possesses several features that differentiate it from traditional networks. The following are SDN’s primary features: Centralized vs. distributed control: In traditional networks, the decision making process, for forwarding data is distributed among network devices. Each device independently uses its routing tables and protocols to make these decisions. However in SDN a centralized controller takes on the responsibility of making decisions regarding network behavior and configuring network devices accordingly. This centralized control allows for enforcement of policies and simplifies network management (Farooq, Riaz & Alvi, 2023).

Network management: SDN architecture simplifies network management by centralizing control and providing a single point of configuration. Using the SDN controller, network administrators can establish and implement policies for the entire network, reducing the complexity of managing individual network devices. This centralized management allows for the allocation, oversight and resolution of network related tasks (Alsaeedi, Mohamad & Al-Roubaiey, 2019).

Network configuration: In traditional networking, network configurations usually revolve around devices and require manual configuration, on each device. However, SDN provides a programmable and policy based approach, to configuring networks. Network administrators have the ability to centrally define network policies and configurations which are subsequently implemented on network devices through the SDN controller (Zoradia & Indumati, 2023).

Flexibility: SDN provides a level of flexibility that is unmatched when it comes to configuring networks and deploying services. Network administrators have the ability to adjust network behavior on the fly and adapt to evolving needs by separating the control plane from the data plane. Network services can be rapidly provisioned and customized through software applications running on the controller, allowing for greater agility and innovation (Kaljic et al., 2019).

Scalability: In traditional networks, scalability becomes a challenge when the complexity of the network increases. Making configuration changes or adding network devices can be time consuming and prone to errors (Kaljic et al., 2019). SDN architecture provides scalability advantages by dividing the control plane from the data plane. Centralized control facilitates the management of large-scale networks more efficiently (Atwal, 2019). Additionally, SDN enables the use of commodity hardware in the data plane, reducing costs and increasing scalability. Scalability of SDN is a concern because one of its main features is centralized control. Among the difficulties presented by a standalone controller are geographic limitations, capacity and performance bottlenecks, and single points of failure. However, it implies that numerous controllers are required to provide short reaction times and good availability when managing a large network, as a single physical controller is insufficient. Thus, having a physically distributed control plane with logical centralization while maintaining the benefits of SDN was a realistic next step forward. One major benefit provided by SDN is the capacity to scale the network dynamically according to demand (Abuarqoub, 2020).

Network visibility and monitoring: Traditional networks often face limitations in terms of their ability to observe and understand network traffic necessitating the use of monitoring devices, for capturing and analyzing traffic. SDN offers improved visibility by means of monitoring and control. The controller is able to collect and analyze flow data and network statistics providing a view of the network that facilitates network monitoring and troubleshooting capabilities (Zoradia & Indumati, 2023).

Vendor independence and interoperability: Traditional networks often rely on protocols and configurations that’re specific to certain vendors. This can create limitations when it comes to interoperability between equipment from vendors. SDN on the another hand aims to promote independence from vendors by standardizing protocols used between the controller and network devices such as OF. This allows for multi-vendor deployments, easier integration of new devices, and more interoperable network solutions (Zoradia & Indumati, 2023).

Network security: Both traditional networks and SDN face security challenges. However, SDN’s centralized control and programmability can help improve network security. SDN allows for fine-grained policy enforcement, rapid response to security incidents, and dynamic adaptation to changing threats. Security policies can be centrally defined and applied consistently across the network (Ahmad & Mir, 2021).

Centralized control, adaptable network management, and scalable deployments are made possible by SDN architecture, which consists of the control plane, application plane, data plane, and SDN controller. The OF protocol facilitates network control and configuration by acting as a standardized communication mechanism between the controller and switches. SDN architecture offers advantages, such as simplified network management, increased flexibility, and improved scalability. These benefits make SDN a transformative approach to networking, with significant potential for enhancing network infrastructures.

SDN vs. traditional network

SDN and traditional networking represents two different methods, for designing and overseeing network architecture. Figure 5 shows the comparison of SDN and traditional network architecture. SDN architecture is different than the traditional network in many aspects, include:

Figure 5 SDN architecture vs. traditional network.

The main differences between the traditional network architecture and SDN architecture as clarified in Table 2.

Table 2 Differences between traditional network and SDN.

Characteristics	SDN	Traditional network	
Network control centralized	✓		
Programmability	✓		
Flexibility of network	High	Low	
Complex control network	Low	Medium	
Network configuration	High	Low	
Management and monitoring	High	Low	
Network security	High	Medium	

It is worth noting that SDN is not intended to completely replace traditional networking but rather to provide an alternative approach with distinct advantages. In practice, organizations may adopt a hybrid approach, leveraging SDN in certain parts of their network while maintaining traditional networking in other areas.

Related study

This section reviews recent studies in the field of SDN, summarizing their key problems regarding SDN security, methodologies used, and what are the possible solutions according to the best of our knowledge presented in Table 3.

Table 3 Existing work in SDN Cybersecurity field.

Ref.	Addressed problems	Solution	Limitations	
Toshniwal et al. (2019)	Lack of initial access permissions assignment to apps.	Behavior-based Access Control, Dynamic Permission Management	Ensuring consistent policy updates across multiple applications running simultaneously is a challenging task that requires careful attention when implementing BEAM.	
Hu et al. (2021)	Malicious apps	API Access Control, Secure Application Management	The scalability of SEAPP is a concern because it does not support various controller platforms. It is also important to examine the security of the controller API.	
Garg et al. (2019)	Lack of intrusion and anomaly detection.	Hybrid Deep Learning, Anomaly Detection	A bottleneck may form when the size of the blockchain grows because OF switches need a lot of memory to store blockchain.	
Tang et al. (2019)	Lack of intrusion and anomaly detection.	GRU-RNN, Intrusion Detection	A network’s performance usually tends to decrease with increasing network size.	
Alshra’a & Seitz (2019)	Lack of packet in message authentication.	Inspector Device, Packet Injection Prevention	A highly complex technique that might affect network performance.	
Imran et al. (2019)	DoS Attacks	DoS Mitigation, Malicious Traffic Blocking	Only identifies attacks against SDN infrastructure. For instance, controller, control channel, and OF switch. It is also unable to identify flow conflicts.	
Sahoo et al. (2020)	DDoS Attacks	KPCA, Evolutionary SVM, DDoS Detection	While the model is effective at identifying attack traffic in environments with a single controller, it might not be able to do so in environments with multiple controllers.	
Aujla et al. (2020)	DDoS Attacks	Blockchain Security, SDN Protection	They address two planes in their solution: the security of data plane and control plane.	
Yazdinejad et al. (2020)	DoS Attacks	Blockchain Security, Packet Parsing	Doesn’t support scalability or smart contracts because BPP operates in the data plane and SDN’s functions and applications operate in the control plane layer.	
Varghese & Muniyal (2021)	Identifying DDoS attacks without using excessive resources.	DPDK framework, IDS Framework, DDoS Detection	Restriction of the framework to a limited number of attacks and certain destination ports, restriction of the mitigation module using particular mitigation techniques and the absence of dynamic resource allocation.	
Eom et al. (2019)	Threats that can reduce the security level of SDN architecture include both new and old ones.	TV-HARM threat modelling	The functional distinctions between the ODL and ONOS controllers were not taken into account in the security analysis, which could have affected the security assessment’s accuracy and the selection of countermeasures.	
Krishnan, Duttagupta & Achuthan (2019)	To mitigate the major security issues in data plane in SDN infrastructure.	Multi-Plane Security, Intrusion Detection	It is necessary to increase the detection accuracies.	
Sebbar et al. (2019)	Current security proposals provide no defense against zero-day attacks or unknown malware (which lacks a signature at the database level of the IPS).	Intrusion Mitigation, Security Algorithm (KPG MT algorithm)	Used manual algorithm to predict and mitigate attacks.	
This SLR article	Lack of comprehensive reviews classifying SDN security risks and systematically analyzing countermeasures.	Conducted a SLR using PRISMA, categorized threats across SDN layers, analyzed countermeasures, and identified research gaps.	Does not propose novel technical solutions	

In the research conducted by Jimenez et al. (2021), authors reviewed the primary security challenges associated with SDN architecture and explore suggested approaches for detecting or mitigating them. The STRIDE threat modeling methodology is applied to all planes and communication channels between them in the SDN architecture which classifies various threats, such as spoofing, tampering, repudiation, information leakage, denial of service, and elevation of privilege. In addition, it provides better response to a changing world of threats. The methodology was picked based on how well-developed and widely used it was for assessing a systems or infrastructure security. This article discussed security of SDN architecture and highlighted unsolved issues. It has been determined that new fault tolerance mechanisms must be developed in order to guarantee availability and consistency, to establish standardized communication interfaces between planes, early detection must be improved, and to develop a new proposal for managing SDN networks security.

In another study authored by Farooq, Riaz & Alvi (2023), authors conducted an SLR by selecting 68 articles to identify security breaches that aim to compromise SDN planes, such as the control, data, and application planes. Next, authors identified the approaches that researchers utilized to create security solutions for SDN planes. After researching and determining the security issues with SDN’s application, control, and data planes, it became evident that these challenges are not limited to a single tier. Consequently, the security concerns within the SDN architecture encompass all planes of the system. Accordingly, they proposed a security solution to address the various security aspects across each plane. In light of this, they proposed an ideal security model to effectively counteract these attacks. Based on the identified security challenges and solutions, the researchers summarized the current security solutions. After that, they identified several gaps and proposed future directions for further investigation, considering these identified gaps.

Bhuiyan et al. (2023) investigated the vulnerabilities exist in SDN architecture with a specific emphasis on the control plane as the primary focal point. The researchers created a comprehensive attack taxonomy that groups many assaults that target the SDN control plane according to their specific attack surfaces. Then, they provided a taxonomical representation of the discovered attacks, providing a systematic framework for understanding and analyzing their characteristics. Give a brief knowledge of various attacks that may targeting SDN control planes. The representation of threats, mitigation strategies, and research gaps analyses, all of them will give a valuable value to the body of knowledge on SDN security.

Rahouti et al. (2022) presented a comprehensive analysis on the fundamental functionality of SDN, focusing on secure communication across various scales. Special attention is given to face the challenges associated with securing SDN-based communications by providing an updated examination of security issues within SDN infrastructure and discussing current practices for enhancing security across each SDN plane. To address a wide range of security threats in SDN architecture, they used a series of techniques covering rootkits, teleportation, flow diversion, authorization, controller placement, network and application manipulation, and topology discovery. Also, SDN systems face security risks at practically every layer of the infrastructure. Furthermore, when the network scale increases, the probability of encountering such attacks also proportionately rises. Conducting a performance analysis of data flow scheduling is essential to ensure a reliable risk assessment. In the context of a secure communication framework, intelligent SDN controllers ought to provide reliable data forwarding and efficiently handle every performance parameter.

Rana, Dhondiyal & Chamoli (2019) studied the advantages of utilizing SDN in contrast to traditional network approaches. It provides an analysis of the SDN infrastructure and its key components, namely the control plane, data plane, and application plane. Furthermore, it addresses the diverse challenges that SDN faces, such as scalability, reliability, and security, and proposes strategies for effectively managing them. The authors also highlight software tools like MININET, RouteFlow, VERIFLOW, and Nettle that can aid in the development of SDN, with proper software development life cycle.

In Zoradia & Indumati (2023), authors conducted a comprehensive exploration of the fundamental distinctions between traditional networks and SDN. It offers an overview of SDN, including its definition, architecture, benefits, and security challenges across each layer. Additionally, it conducts a thorough examination of the design of the SDN networking paradigm, reevaluating previous efforts undertaken to address challenges such as performance, security, scalability, and reliability.

BEhavior-based Access control Mechanism (BEAM) has been proposed by Toshniwal et al. (2019) where they utilized network behavior measures, such as rate of flow injection or packet in rate, obtained from an IDS to assign permissions to third-party applications. The permissions of the applications are dynamically adjusted in real-time based on the analysis performed by the IDS. The elements that make up BEAM are as follows: a policy engine that creates policies for granting or denying application permissions, two essential databases, a policy store and a mapping table and a registration handler that handles initial application registration and permission assignment. The activity engine and activity detection module in BEAM also evaluate application activity using logs and IDS data, respectively.

SEAPP was suggested by a research article Hu et al. (2021) presenting a structure with two elements. The initial part is the permissions detection engine tasked with verifying the credibility of application permissions. Meanwhile, the second element, referred to as the registration authorization engine, manages the app’s authentication and registration procedures. To enhance security and safeguard against tampering or eavesdropping threats, the framework integrates the Number Theory Research Unit (NTRU) algorithm.

Badotra & Panda (2020) conducted a study comparing the security measures of two drivers: OpenDayLight (ODL) and Open Networking Operating System (ONOS). Their analysis revealed that while no controller can be entirely secure from attacks, ODL exhibited robust defenses against security threats compared to ONOS, which ranked as the second-best controller in terms of security measures.

Garg et al. (2019) introduced an approach that employs deep learning to identify abnormal occurrences effectively. The framework comprises two elements: the anomaly detection module and the data delivery module. Emphasizing security as its concern in the initial module allows for addressing a range of potential attack vectors, like malware infections and attempts at spoofing or hijacking. To gather and classify flow characteristics that are exchanged between the controller and network elements, it makes use of the restricted Boltzmann machine (RBM) and support vector machine (SVM) algorithms. By analyzing these features, the system generates an anomaly report that allows the controller to respond effectively by managing deletion and communication to mitigate detected anomalies.

Tang et al. (2019) presented a method for intrusion detection systems using deep learning. Their approach comprises three modules: an anomaly detector to identify unusual patterns in network traffic data; an anomaly mitigator to address detected anomalies; and a flow collector that collects important information using packets in messages from the network traffic flow stream. In this setup, the anomaly detector module utilizes a gated recurrent unit recurrent neural network (GRU RNN) for detecting anomalies. Lastly, the anomaly mitigator module determines if the traffic should be ignored or thoroughly examined.

Alshra’a & Seitz (2019) introduced a hardware-based solution to address packet injection attacks. The proposed solution entails incorporating a device into the network infrastructure. This device uses a database of approved hosts by referring to packet-in messages to confirm their legitimacy. If a host is unable to authenticate, its packet-in message is returned as invalid.

Detection And mItigation SYstem (DAISY) has been proposed by Imran et al. (2019) where authors aimed to identify and mitigate denial of service (DoS) attacks. DAISY contains of four primary functions. Information from packet in messages is gathered and stored by the data collection function. Statistical analysis is employed by the threat detection function to detect requests that are excessively made from a particular host. When this kind of excessive activity is found, the traffic is flagged as suspicious and is temporarily blocked by the attack prevention function. If the host persist in sending requests at the same pattern, the traffic will be flagged as a malicious and will be blocked permanently. Finally, after every system iteration, the threat value reduction function lowers the threat level by updating the blocking flow rules.

Sahoo et al. (2020) presented a machine learning (ML) framework for the detection and mitigation of DDoS attacks. There are several modules in this framework. First, switches provide flow statistics data to the statistics monitor module, which periodically gathers this data. After that, to extract flow characteristics from the gathered data, the feature extractor module applies kernel principal component analysis (KPCA). The classifier module uses the SVM classifier with parameter optimization through the genetic algorithm (GA) after these features are extracted. This procedure makes it possible to distinguish between malicious and benign traffic. Ultimately, this data is used by the mitigation module to create a rule that permits the underlying network to reject the malicious traffic that has been identified.

Aujla et al. (2020) presented a Blockchain (BC) as a service architecture to mitigate DDoS attacks. It focuses on confirming switches’ identities prior to allowing them to access the network for flow transmission. The solution generates public and private keys using BC technology, which are then distributed among the network switches. As a result, when a device makes a request on the network, the other BC members decide whether to approve or reject the transaction based on their consensus.

Yazdinejad et al. (2020) proposed the blockchain enabled packet parser (BPP), which is a novel architecture aim to detect various types of attacks, with a focus on DoS attacks. Programming Protocol-Independent Packet Processors (P4) and BC technology are combined in this solution. P4 allows for packet processing within the data plane, and the BC component examines packet behavior. The controller is notified when an attack is detected, and pre-established policies are used to take the necessary action.

The DPDK-based DDoS Detection (D3) framework has been proposed by Varghese & Muniyal (2021) to handle IDS performance problems as well as DDoS attack-related design concerns in SDN. This framework incorporates intelligence in the data layer by means of the Data Plane Development Kit (DPDK) in the SDN architecture. Because of its fast packet processing and data plane monitoring characteristics, DPDK is suitable for this usage. The D3 framework resolves the conflict between DDoS Attack and SDN architecture as well as the IDS constraint in high-speed networks. Furthermore, the D3 detection system provides a dependable assault prediction with good detection performance. By using DPDK in the data plane to create a statistical anomaly detection method as a virtual network function (VNF), D3 enables rapid detection of DDoS attacks. The statistical anomaly detection algorithm’s capacity to identify DDoS attacks is confirmed by the publicly available CIC-DoS statistics, and experimental findings support the effectiveness and efficiency of the D3 framework. The framework is cost-effective because it doesn’t need any extra hardware to function.

The Threat Vector Hierarchical Attack Representation Model (TV-HARM) has been proposed by Eom et al. (2019), which is a novel graphical security model formalism that was developed by combining various threat vectors. It provides a structured approach to evaluating threats, attacks, and countermeasures for SDN. Three tasks carried out by TV-HARM enable SDN security risk assessment: (1) record dynamic SDN changes, (2) assess intricate attack and defensive scenarios and (3) use a variety of SDN security metrics to describe the security posture. Additionally, they used three security evaluation criteria to evaluate the security of SDN: attack effect metrics, vulnerability scores, and network centrality metrics. The results demonstrated how well the proposed approach evaluated SDN security by accounting for a variety of SDN threat vectors.

In Krishnan, Duttagupta & Achuthan (2019), an adVanced multi-plAne secuRity fraMework for softwAre defined Networks (VARMAN) has been proposed as a network security and intrusion detection system (NIDS) for SDN. Coarse-grained flow monitoring algorithms are incorporated into this SDN security scheme on the dataplane to promptly identify and anticipate network-centric DDoS/botnet attacks. On the control plane, a fine-grained hybrid deep-learning based classifier pipeline is also put into practice. The hybrid model is an enhanced SDN stack that integrates both shallow and deep learning methods. The dataplane includes mechanisms for feature selection, effective data filtering, behavioral triggers, attack prediction, and data reduction methods. The scientists refined well-known datasets like NSL-KDD and CICIDS2017 to produce a balanced dataset with an equal proportion of malware samples and normal traffic. The outcomes show that VARMAN is very fast and accurate at identifying and reducing security threats in SDN environments.

Sebbar et al. (2019) presented an assessment of the security framework in place. Additionally, they demonstrated an algorithm called KPG-MT, which summarizes the primary security techniques used to mitigate various threats and detect, analyze, and eliminate malicious SDN nodes. Furthermore, in order to demonstrate the effectiveness of the suggested framework, they suggested putting this algorithm into practice at the level of several locations (within the controller, on control plans, and on distributed control plans) and employing various attack scenarios (man in the middle, DoS, and malware-based attacks). Their suggested algorithm’s primary objective is to assess the current security frameworks.

This article offers several key advantages compared to prior studies: Comprehensive coverage: The review goes beyond existing research, which concentrates on individual SDN layers or attack types, to provide a complete assessment of threats across all SDN planes (control, data, application) and interfaces (northbound, southbound, east/westbound).

Methodological rigor: The PRISMA framework is used for systematic literature review to maintain transparency and reduce bias when selecting articles and conducting analysis.

Taxonomy and classification: A structured taxonomy of SDN-specific attacks, vulnerabilities, and countermeasures is developed to enhance comprehension and comparison.

Future directions: This study pinpoints essential research gaps (including interface standardization and scalability of blockchain/AI solutions) and suggests practical directions for future work that existing research frequently fails to address.

Practical insights: The article brings together different methodologies like machine learning and blockchain and assesses their applicability to provide practitioners with a useful reference for building secure SDN architectures.

The inclusion of these elements closes the gap between separate studies and creates an essential foundation for both SDN security research and deployment.

Discussion

The findings of this article strongly contribute to SDN security studies by creating a comprehensive method to present threats and countermeasures across all SDN planes and interfaces that fill holes found in the current literature. This review employs the PRISMA methodology for unbiased literature analysis to offer consistent security challenge insights alongside the reason for utilizing blockchain and AI within SDN networks for more robust security. The taxonomy with analysis documented in this article creates a foundation that supports researchers and practitioners to build better security architectures. The study finds its position within existing research to enhance its impact on future SDN security framework advancements.

Even though numerous research studies have examined SDN security to identify vulnerabilities and possible countermeasures along with opportunities for improvement, still fundamental challenges exist, particularly in regard to control plane centralization along with attack surface expansion and integration complexities with emerging technologies. The section includes a systematic comparison of current research studies to reveal security gaps and rate the suggested solutions while suggesting directions for future work to enhance SDN security architectures. Discussion includes ordinary threats such as DDoS attacks against control planes together with controller hijacking and data plane manipulation attacks, along with more specific issues like establishing trust among distributed controllers and scaling blockchain security models. This section examines the effectiveness of current security measures, which include machine learning-based anomaly detection and blockchain authentication together with programmable access control, and it seeks to deliver a more detailed understanding of their capabilities and constraints to showcase the direction for future progress. 1. Security challenges in SDN: Throughout the previous research articles, a common thread emerges regarding the discovery of security weaknesses that are built into the architecture of SDN, most notably in the separation between the control and data planes. Researchers in Bhuiyan et al. (2023) highlighted vulnerabilities within the control plane and discussed various types of attacks that take advantage of these weaknesses. Likewise, Rahouti et al. (2022) and Rana, Dhondiyal & Chamoli (2019) focused on these vulnerabilities and explored ways to address them. Also, they highlighted that there are risks in the control infrastructure that could lead to issues like DoS along, with forms of attacks aimed at the controller.

Researchers widely agree on the significance of securing the control plane due to its vulnerability consistently noted in studies. However, some research studies, like those conducted by Alshra’a & Seitz (2019) also explore threats in the data plane, such as injection attacks. Expanding the focus of SDN security research beyond the control plane, such studies add depth to the understanding of potential vulnerabilities. Table 4 shows some security issues and solutions from the previous studies concisely.

2. Methodologies: Various approaches have been suggested to tackle these security concerns.

Various research studies by Garg et al. (2019) as well as Tang et al. (2019) utilize ML and deep learning (DL) techniques to identify and address anomalies within SDN. These methodologies provide the flexibility for systems to continuously learn from network traffic behaviors and identify abnormalities as they happen in time. However, the increase in performance requirements can pose challenges when dealing with larger and more intricate networks.

Using blockchain to secure SDN brings about logging and distributed control mechanisms that boost security by guaranteeing the genuineness and accuracy of information shared between controllers and switches (Aujla et al., 2020; Yazdinejad et al., 2020). However, the scalability of blockchain-driven solutions poses a hurdle, particularly in bigger SDN scenarios.

Novel approaches like behavior-based access control (BEAM) and SEAPP have been suggested to handle security at the application level within SDN. BEAM focuses on granting permissions to applications according to network behavior as proposed in Toshniwal et al. (2019) while SEAPP leverages the NTRU encryption algorithm for boosting application-level security (Hu et al., 2021). These methods encounter challenges in maintaining uniformity across various applications and aligning with current SDN controllers. Table 5 summarizes the methodologies that have been used in some literature above.

3. Countermeasures effectiveness: While there are different solutions proposed, their effectiveness varies. Deep and machine learning models are well regarded for their flexibility and ability to detect anomalies in time. However, there are worries about their resource usage and the occurrence of false positive comments. As network traffic expands, AI algorithms may encounter difficulties with performance necessitating tuning and improvements. While blockchain provides increased transparency and security benefits, its implementation in paced SDN brings about challenges related to delays and scalability. Even though innovative approaches such as BPP (Yazdinejad et al., 2020) offer the potential to identify DDoS attacks, they might face difficulties when processing amounts of data instantaneously. Alshra’a & Seitz (2019) suggested a hardware-based approach to combat injection attacks. While providing security measures, this solution may lead to higher network intricacy and potentially impact performance levels.

4. Limitations: While the methodologies that were used in previous studies have their merits, the studies do recognize constraints. Scalability is a concern in various solutions like blockchain technology and machine learning when applied to extensive SDN. Research shows that methods successful in networks may not work as effectively in larger ones. Numerous research works highlight the absence of uniform security standards for SDN interfaces, such as southbound APIs leading to challenges in integrating various security measures effectively. AI-driven and blockchain-based solutions tend to use up a lot of resources, as they often need more computing power and bandwidth than usual. This increased demand could potentially affect the performance of networks in applications that require real-time processing. Table 6 summarizes The existing limitations after applying some countermeasures.

Table 4 Security challenges in SDN.

Ref.	Challenge/Solutin	Layer/Plane	
Bhuiyan et al. (2023)	Threats in the control plane	Control	
Rahouti et al. (2022)	Security issues in all SDN planes, also focusing on securing communications.	Control, Data, Application	
Rana, Dhondiyal & Chamoli (2019)	Security, Scalability and reliability issues in SDN.	Control, Data	
Alshra’a & Seitz (2019)	Packet injection attacks	Data	
Garg et al. (2019)	DDoS attacks detection using machine learning models	Data and Control	
Tang et al. (2019)	Anomaly detection using deep learning models	Control	
Yazdinejad et al. (2020)	DDoS attacks detection using Blockchain and P4 programming	Data	
Aujla et al. (2020)	DDoS attacks mitigation using blockchain as a service	Data and Control	

Table 5 Used methodologies in some previous studies.

Ref.	Methodology	Technology/Tools used	
Bhuiyan et al. (2023)	Taxonomy of attacks	STRIDE threat modeling	
Rahouti et al. (2022)	Comprehensive security analysis	Rootkits, flow diversion, topology discovery	
Rana, Dhondiyal & Chamoli (2019)	Analysis of SDN components	MININET, RouteFlow	
Alshra’a & Seitz (2019)	Hardware-based packet injection solution	Independent hardware implementation	
Garg et al. (2019)	Deep and machine learning for anomaly detection	Restricted Boltzmann Machine (RBM), Support Vector Machine (SVM)	
Tang et al. (2019)	Intrusion detection using deep learning models	Gated Recurrent Unit (GRU-RNN)	
Yazdinejad et al. (2020)	DDoS attacks detection system using Blockchain	Blockchain, P4 programming	
Aujla et al. (2020)	Securing network using blockchain as a service	Blockchain	

Table 6 The existing limitations after applying some countermeasures.

Ref.	Countermeasures	Attack targeted	Limitations	
Bhuiyan et al. (2023)	Fault tolerance, improved early detection	Control plane attacks	Difficulty in ensuring availability and consistency after DoS attacks	
Rahouti et al. (2022)	Secure communication techniques across SDN	Network communication attacks	Potential challenges in large-scale secure communication	
Rana, Dhondiyal & Chamoli (2019)	Reliability and security mechanisms	Control and Data plane issues	Manual processes, lack of automation	
Alshra’a & Seitz (2019)	Hardware-based packet validation	Packet injection	High complexity in hardware-based approaches	
Garg et al. (2019)	AI-based anomaly detection	DDoS attacks	High computational overhead in large-scale networks	
Tang et al. (2019)	AI-based anomaly mitigation system	Anomalies in traffic	AI models may struggle in large environments	
Yazdinejad et al. (2020)	Blockchain for packet parsing and validation	DoS attacks	Scalability issues with blockchain growth	
Aujla et al. (2020)	Blockchain for secure switch authentication	DDoS attacks	Bottleneck when blockchain grows	

Security challenges and countermeasures in SDN

SDN offers numerous benefits, but it also introduces unique security challenges and vulnerabilities. Attackers can take advantage of security vulnerabilities in the SDN planes by employing several kinds of attacks that specifically target these vulnerabilities. The SDN architecture’s communication channels are one example of vulnerabilities that could be used by adversaries to obtain access to the controller and do malicious actions, and compromise the security of the network. It is important to explore the countermeasures that will be used to address or mitigate the security issues in the SDN architecture. The challenges and countermeasures of SDN architecture were analyzed in this section. In addition, the SDN architecture and its communication channels are divided into three blocks: the east/west interface and control plane, the southbound interface and data plane, and the northbound interface and application plane. Table 7 lists vulnerabilities, attack types that could target planes and interfaces between them, and countermeasures to be used to address those types of attack.

Table 7 Security challenges and countermeasures in SDN.

SDN layer	Types of attacks	Vulnerabilities	Countermeasures	
Application plane north-bound interface	Diversity of the SDN applications and their implementation by the third parties which can lead to impersonation attack, access the controller and make changes to the configuration.

Lack of effective authentication and access control mechanisms.

	Information manipulation, impersonation, eavesdropping or tampering attacks, information disclosure.

	Proper authentication and access control mechanisms like coarse and fine grained access controls, RBAC, MAC and DAC.

Integrating BC with access control mechanisms (Hoang, Duy & Pham, 2019).

Analyze the behavior of the network by using IDS (Toshniwal et al., 2019).

Using encryption methods (Hu et al., 2021).

	
Control plane east/West-bound interface	Dependence on a single control in a centralized controller may result to a single point of failure in the network.

A lack of trust, authentication procedures, and encryption between distributed controllers.

Absence of provenance verification for the data sent across controllers in the east/westbound interface.

A controller shutdown may result from a large frames registration.

	DDoS, zero-day vulnerabilities, repudiation, impersonation, hijacking, tampering, and information disclosure.

	Identity-Based Cryptography (IBC) should be used to secure communication at the east/westbound interface.

Using secure controllers like ODL or ONOS.

Using appropriate authentication mechanisms between distributed controller.

Using IDS.

Integrating emerging technologies like DL, ML, BC with the traditional technologies like IDS, IPS, authentication mechanisms and encryption.

	
Data plane south-bound interface	Misconfiguration of TLS protocol.

From the client side, lack of use of certificates during handshake time in the authentication process.

The absence of authentication mechanisms.

Inadequate authentication mechanisms to confirm the origin of LLDP packet.

The flow tables are overloaded.

	MitM, DoS, hijacking, fabrication, and DoS (LLDP flooding or packet injection).

	Using IBC protocol to protect communications and the data plane in the southbound interface (Yigit et al., 2019).

The overhead of communicating between the data and control planes should be reduced by using statefull data plane (Almaini et al., 2019).

Observing the OF messages exchanged between the switch and the controller.

Detecting anomalous activity by comparing the LLDP message payload size with the Maximum Transmission Unit (MTU) size that is permitted.

	

Major security challenges and vulnerabilities in SDN

SDN provides adaptability and customization options alongside management but comes with distinct security risks and weaknesses as well. The main security concerns and vulnerabilities in SDNs stem from the segregation of control and data functions within the network architecture and the ability to program the network settings through an SDN controller. Control plane attacks: Attackers might focus on the control plane of an SDN environment in order to disrupt or manipulate the behavior of the network. These attacks on the control plane can involve compromising the SDN controller taking advantage of vulnerabilities in the controller software, or intercepting and tampering with the control messages exchanged between the controller and switches (Bhuiyan et al., 2023). A controller that has been compromised or is acting maliciously can cause harm. When a controller is, in the hands of an actor they might send out control instructions that are intended to cause network behavior intercept network traffic or disrupt service. Furthermore if a controller becomes compromised it could manipulate network policies compromise the segmentation of the network or assist attackers in moving within the system (Bhuiyan et al., 2023). SDN controllers play a vital role in network operations security. A breach in this component could result in security risks affecting the entire network (Kreutz, Ramos & Verissimo, 2013; Scott-Hayward, Natarajan & Sezer, 2015). Single point of failure: When the centralized control plane is compromised by the attacker, it opens up a significant vulnerability risk as it could potentially manipulate the entire network.

Denial of Service (DoS) Attacks: Commonly known as DoS, can make the network inaccessible by flooding the controller with data traffic, rendering it inoperable.

Unauthorized access: A compromised controller may grant entry to network assets, which enables attackers to control traffic flow.

Tampering with flow rules: Cyber attackers have the ability to modify the flow rules transmitted by the controller to switches, which may result in manipulating network traffic and redirecting data without authorization.

Data plane manipulation: The data plane is vulnerable to attacks as it handles the task of routing packets according to the instructions provided by the control plane. Attackers have the ability to manipulate the data plane, which can result in access or the extraction of data, by modifying or redirecting network traffic. Attackers might take advantage of weaknesses in the switches. Manipulate the controllers configured forwarding rules compromising their integrity (Vadivu & Rajagopalan, 2023; Shaghaghi et al., 2020). Packet injection attacks: Packet injection attacks occur when malicious packets are inserted into the network to take advantage of the forwarding mechanisms within the data plane. If these packets are not verified correctly by the controller, it could potentially put the network at risk of compromise.

Packet modification and forgery: Cyber attackers can alter existing packets or create ones illegitimately, which could result in unauthorized entry into systems and potential breaches such as data leaks.

Flow table exhaustion: Flow table overload issue occurs when hackers overwhelm the data plane switches with several flow entries, resulting in the exhaustion of the switch flow table and potentially causing performance issues or network breakdowns.

Northbound interface vulnerabilities: The connection going north links the SDN controller with applications operating above the SDN system to facilitate their communication and interaction (Rauf et al., 2021). Application manipulation: Harmful apps could take advantage of weaknesses in the northbound interface to influence the controller’s actions and make alterations to network settings.

Lack of authentication and access control: Weak access control systems within the northbound API may potentially enable malicious actors to run privileged commands that could modify network settings or obtain confidential information.

Impersonation attacks: Impersonation incidents occur when attackers impersonate legitimate apps or services to gain unauthorized entry into SDN control functions.

Southbound interface vulnerabilities: The connection going south from the SDN controller links up with the network’s base infrastructure, like switches and routers, mostly using OpenFlow protocol that carries its set of security vulnerabilities (Tupakula et al., 2022). Lack of encryption in communication: Communication between the controller and the switches through the interface could be at risk of being intercepted or altered without proper encryption measures, like TLS, in place.

Man-in-the-Middle (MitM) attacks: Cyber attackers might carry out MitM attacks to intercept communication between the controller and switches, which could lead to altering commands or injecting instructions.

Protocol misconfigurations: Mistakes in setting up protocols like OpenFlow can make the network susceptible to security threats.

East-West interface vulnerabilities The east/westbound interface uses the Eastbound API to enable the connectivity of distributed controllers. One of the main features of SDN is centralized network control to provide efficient coordination across multiple controllers (Latif et al., 2020). On the other hand, this interface is utilized for load balancing and redundancy but contains some security vulnerabilities. Inter-controller trust issues: Issues with trust between controllers can arise when communication between them is not properly protected. If one controller is compromised, it could have an effect and impact other controllers in the network, resulting in a widespread compromise of the entire system.

Lack of provenance verification: Proper verification of the source is crucial as it prevents parties from inserting false data into the system, which could result in inaccurate routing or flow of traffic within the network.

Controller overload: SDNs are in charge of handling network tasks. However, they are also vulnerable to being targeted by attacks (Hamdan et al., 2021). DoS attacks: This type of attack uses traffic to overload its capability to manage genuine traffic and ultimately leads to network downtime.

Flow rule overloading: Attackers have the ability to take advantage of weaknesses in the flow rule management system to overwhelm the controller with an abundance of flow rule requests. This can result in a decrease in performance or even system failure.

Lack of standardized security frameworks: A major obstacle to SDN security is the lack of established security frameworks and protocols that are standardized across the board (Jimenez et al., 2021). Inconsistent security measures across vendors: SDNs pose a challenge in implementing network-wide security policies due to differences in security features offered by each vendor.

Interface standardization issues: When there are no standards for the interfaces going north and southbound in the system, it can be tricky to ensure secure communication between the SDN controller, the applications it runs, and the network infrastructure underneath.

Implications on information security, network availability, and data confidentiality

Unauthorized entry, into the SDN infrastructure, which includes the controller, switches, and network applications can lead to security breaches. Individuals who gain access have the ability to manipulate network settings listen in on data transmission launch attacks, within the infrastructure, or compromise the confidentiality, integrity, and availability of network management functions (Toshniwal et al., 2019). Here, the implications of attacks on information security, network availability, and data confidentiality were discussed. Then, Table 8 provides an example of each situation.

Table 8 Implications on information security, network availability, and data confidentiality.

Challenge	Information security	Network availability	Data confidentiality	
Control plane attacks	Tampering with the data in the control plane can compromise the functioning of network management.	Compromised controllers have the potential to lead to service disruptions on a scale.	Hackers could potentially access information from the control plane and compromise its confidentiality.	
Data plane manipulation	Tampering with packets through injection can change data and compromise the integrity of network communication.	Manipulating flow rules could result in traffic interruptions or redirections that may impact the performance of the network.	Unauthorized packet forwarding has led to data being stolen by cybercriminals.	
Northbound interface vulnerabilities	Inadequate authentication enables apps to alter network regulations and undermine integrity.	Issues with network connectivity arise from misconfigures caused by compromised applications.	Unauthorized access to data can occur as a result of impersonation attacks.	
Southbound interface vulnerabilities	MitM attacks targeting the connection can enable malicious actors to modify or intercept data transmissions and compromise security measures in place.	Attacks that focus on OpenFlow have the potential to interrupt communication between the controller and switches, which could affect their availability.	Southbound communication can be intercepted and may result in security breaches of data.	
East-west interface vulnerabilities	Compromised communication between controllers could put network security at risk.	Failure in communication between distributed controllers can lead to network fragmentation.	Sensitive data that is shared among controllers has the potential to be revealed by the attackers, which impacts the data confidentiality.	
Controller overload	Controllers that are overwhelmed could fail to handle traffic flow on networks, which may result in security vulnerabilities occurring.	Issues with service interruptions or reduced quality are caused by an overload in the controller.	Mismanaged traffic may expose sensitive information during high-load conditions.	

Information security: Security issues in SDN can give rise to entry into data, interception of information and unauthorized alteration of network settings. These risks can jeopardize the confidentiality, integrity and availability of network traffic potentially resulting in data breaches and unauthorized exposure of information (Deb & Roy, 2022).

Network availability: Attacks that focus on the control or data plane have the potential to disrupt network availability resulting in service outages or decreased performance. When the SDN controller is compromised or when forwarding behavior is manipulated attackers can create network congestion redirect traffic to existent destinations or launch DoS attacks. These actions negatively impact the availability and overall performance of network services (Deb & Roy, 2022).

Data confidentiality: The security of data can be at risk if unauthorized individuals gain access to network traffic or manipulate the data flow. When sensitive information is transmitted through the network there is a possibility that it could be intercepted which may result in data breaches and hence exposure of sensitive data or even compromise encryption keys (Rahouti et al., 2022).

SDN introduces unique security challenges and vulnerabilities, these include attacks, on the control plane, manipulation of the data plane, unauthorized access and malicious behavior by controllers. It is crucial to address these security concerns in SDN environments to safeguard information security, maintain network availability, and preserve data confidentiality. Real life examples and case studies highlight the significance of tackling these issues to ensure the resilience and integrity of SDN systems.

Countermeasures and security solutions

Addressing security challenges in SDN requires the implementation of robust security mechanisms and solutions. This section provides a comprehensive analysis of existing security measures, including traditional security measures and emerging technologies along with their effectiveness in mitigating the identified security challenges in SDN. Access controls: to ensure that authorized entities can access and make changes to the SDN infrastructure. It is beneficial to implement access controls, like attribute based access control (ABAC) and role based access control (RBAC). These access controls can be applied at the controller, switch and application levels to safeguard against access and control plane attacks (Hu et al., 2021).

Authentication: implementing robust authentication methods, such, as authentication between the controller and switches plays a key role in safeguarding the SDN infrastructure against unauthorized access. Employing techniques, like certificates, multifactor authentication and secure exchange protocols enhance the overall authentication process (Hu et al., 2021).

Encryption: securing the confidentiality and integrity of information transmitted within the SDN environment involves encrypting the control and data plane traffic. One way to achieve this is, by implementing techniques like TLS or Internet Protocol Security (IPsec) which can be applied to establish communication channels between the controller and switches and switches themselves (Hu et al., 2021).

Firewalling: installing firewalls, within the SDN infrastructure is crucial, for filtering and overseeing network traffic safeguarding against access and defending against a range of attacks. Firewalls serve the purpose of upholding security policies identifying irregularities and offering a layer of protection (Hu et al., 2021).

Traditional security measures, such as access controls, authentication, encryption, and firewalling, can be applied in SDN environments (Chica, Imbachi & Vega, 2020). However, the dynamic nature of SDN requires adaptations and extensions to traditional security mechanisms. For example: Access controls should consider the changing nature of network flows and the ability to program network policies.

Authentication methods should take into account the nature of SDN controllers and switches to ensure the safe transmission of control messages.

Encryption methods need to consider the scalability and performance demands of SDN along, with the distribution of encryption keys.

Firewalls should have the ability to analyze and control network traffic at the application layer considering the changing network behavior in SDN.

Here, the significant role of emerging technologies in protecting SDN architecture is presented. Artificial Intelligence (AI): AI has the potential to enhance SDN security by offering threat detection, anomaly detection and behavioral analysis. By analyzing network traffic patterns in real time, AI algorithms can identify any deviations. Promptly detect possible security breaches (Alhaj & Dutta, 2022).

Blockchain: BC technology has the potential to improve security in SDN by introducing logging mechanisms that are both tamper proof and decentralized. Its application can ensure that control plane messages remain authentic and their integrity intact while also enabling the detection and mitigation of any manipulations of the control plane (Choudhary & Dorle, 2022).

Software defined security (SDSec): SDSec aims to incorporate security features into the SDN infrastructure. This entails utilizing security controllers, security applications and security service chaining to implement and adapt security policies in response, to security events (Sallam, Refaey & Shami, 2019).

Intent-based networking (IBN): IBNs main objective is to ensure that the networks actions are in line with business and security policies at a level. It achieves this by converting high level intentions into network configurations. This approach allows for a policy oriented approach to secure SDN, which can result in improved management and implementation of security policies (Sallam, Refaey & Shami, 2019).

Future directions and open research challenges

SDN has emerged as a powerful paradigm for network management and control, offering flexibility, programmability, and centralized control over network resources. However, the increasing adoption of SDN also brings forth various security challenges that need to be addressed to ensure the CIA of network infrastructure and services.

In this section, the potential future directions for securing SDN are explored, taking into account emerging trends, technologies, and industry advancements. These directions aim to enhance the security posture of SDN deployments and enable organizations to build resilient and trustworthy networks in the face of evolving cyber threats. By understanding and proactively addressing these future directions, stakeholders in the SDN ecosystem can stay ahead of potential security risks and ensure the continued success and adoption of SDN technology. Let’s delve into the key areas that hold promise for the future of secure SDN.

Future directions for secure SDN

Some critical security issues with SDN architectures still require thorough examination through additional research. Scalability and performance trade-offs in security solutions: While existing SDN security frameworks—such as blockchain-based authentication systems and AI-driven anomaly detection tools—enhance protection, they often impose significant latency and computational burdens. To address these trade-offs, future work should prioritize the development of lightweight cryptographic protocols, optimized blockchain consensus algorithms, and decentralized AI architectures. These innovations must achieve robust security guarantees while preserving the operational efficiency of the network infrastructure. Advancing this balance will be critical for deploying scalable, high-performance SDN solutions in latency-sensitive environments (Chica, Imbachi & Vega, 2020).

Enhancing threat intelligence and adaptive security: Present security solutions are dependent on static block-based technologies or classic learning machine approaches that cannot adapt to real-time threats. There is a need for research in self-learning AI models, federated learning for distributed IDS/IPS, and real-time threat intelligence sharing frameworks to enhance automated attack detection and mitigation (Alashhab et al., 2024).

Standardization of security frameworks and interoperability issues: The absence of standardized security protocols for SDN controllers and APIs is a problem that complicates security enforcement. Future work should focus on further exploring and proposing vendor-neutral, standardized security frameworks that provide seamless interoperability while securing diverse SDN deployments (Hu et al., 2021).

Quantum-resilient security mechanisms: With the rise of quantum computing, RSA and ECC may become traditional encryption schemes. Research is needed to explore how to integrate quantum-resistant cryptographic algorithms into SDN architectures for future-proof network security (Saritha, Reddy & Babu, 2022).

Trust and authentication in distributed SDN environments: A problem is that in multi-controller SDN architectures, trust among controllers is difficult to establish, particularly if East-West APIs are used. Future studies should develop decentralized authentication frameworks based on Zero-Trust security models and identity management using blockchain (Azab et al., 2019).

Self-healing and autonomous security for SDN: For future SDN security, we should move towards autonomous self-healing systems that can identify, counter, and recover from attacks without human intervention. This entails evolution in the areas of AI-driven predictive analytics, automated threat response, and intent-based security policies (Oladosu et al., 2021).

Software-defined security services: Moving forward one of the areas that holds promise is the creation of security services that can be customized to suit network conditions and evolving threats. This entails incorporating security measures into the SDN infrastructure enabling the provisioning and adjustment of security policies and controls in response, to network context and security needs (Zarca et al., 2019).

Automated security orchestration: In the future advancements, SDN need to focus on automating security orchestration and response. This entails integrating security incident response systems with SDN controllers to enable automated actions, for threat mitigation and remediation. Utilizing decision making algorithms and policy driven approaches can facilitate security responses throughout the SDN infrastructure (Bringhenti et al., 2019).

To achieve this, the gaps in research presented in this article must be addressed; this will strengthen SDN security, promote its adoption in critical infrastructure such as 5G, IoT, and the cloud, and offer a future research direction in this domain.

Open research challenges in secure SDN

Many studies have already been conducted to address different scenarios and security challenges related to SDN architecture. Still, there are many unsolved issues related to SDN architecture that need to be solved. This section discovers open research challenges to secure SDN that need further attention.

Because of the dynamic nature of the SDN architecture, many authors focus on authorization issues in the SDN application plane and emphasize the importance of granularity in permissions (Toshniwal et al., 2019). Also, being the main problem is the trust between applications and the controller, a new research opportunity emerges here to establish a standardized policy that verifies the levels of security in SDN applications. In the control plane, fault tolerance mechanisms are implemented by distributed controllers to recover the service after failure of the system, such as a DoS attack. However, there is disagreement among many authors about how to achieve the balance between consistency and resilience (Ahmad & Mir, 2021). According to Brewer’s theorem, it is not possible to guarantee both availability and consistency after a partition in distributed environments. Although ODL and ONOS controllers use the RAFT algorithm for fault tolerance or strong consistency, there are still gaps to fill in terms of improving its performance and preserving information between controllers after a service interruption. The researchers have to take an action regarding this point from the security perspective. At the data plane, the split and distribution technique adds risk to flow rules because they can overload a node, be lost, or duplicated while propagating which lead to create a new research opportunity.

In addition, most authors focus on addressing attacks that may target the planes of the SDN architecture while ignoring security issues related to interfaces. However, the possibility of attack scenarios will increase in the absence of standardized interfaces. Therefore, standardizing interfaces is considered as the most significant challenges for the security of SDN architecture.

Addressing these research challenges will help advance the state of secure SDN and ensure that future SDN deployments are resilient, scalable, and capable of defending against emerging threats. Collaboration between academia, industry, and standardization bodies is essential to drive research and innovation in these areas.

Conclusions

SDN is a transformative technology with many advantages, including more programmability, scalability, and flexibility in network administration. SDN enables the implementation of policies and centralized control, for network configuration and management by separating the data plane from the control plane. In this study, different aspects of SDN were examined, which include its elements like the SDN controller as well as the SDN planes and interfaces. In addition, the benefits of SDN compared to traditional networks are explored, including increased agility, simplified management, and faster innovation. Additionally, the significance of security in SDN networks and interfaces was emphasized. As SDN networks evolve further it is vital to consider strategies for enhancing their security. These strategies involve integrating threat intelligence and utilizing AI techniques and best practices. By addressing these future directions, the SDN community can enhance the security of SDN networks mitigating emerging threats and ensuring network operations. To sum up, SDN shows potential in transforming network architecture and management by offering flexibility, scalability and security. As technology progresses it is essential to keep up with emerging trends and adopted practices to fully capitalize on the advantages of SDN while maintaining a robust security framework.

Additional Information and Declarations

Competing Interests

The authors declare there are no competing interests.

Author Contributions

Ahmed Aljughaiman conceived and designed the experiments, performed the experiments, analyzed the data, prepared figures and/or tables, authored or reviewed drafts of the article, and approved the final draft.

Seetah Almarri conceived and designed the experiments, performed the experiments, analyzed the data, prepared figures and/or tables, authored or reviewed drafts of the article, and approved the final draft.

Data Availability

The following information was supplied regarding data availability:

This is a literature review.

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
