# Peer review of "The pivotal role of software defined networks to safeguard against cyber attacks: a comprehensive review"

_PeerJ Computer Science, doi:10.7717/peerj-cs.2814_

## Round 0.1 · original submission · Major Revisions

The manuscript is provides valuable insights into SDN and network security. However, addressing the issues raised by the reviewers, refining the paper organization, enhancing the analysis of findings, and making grammar and formatting adjustments will significantly improve the clarity, readability, and impact of the paper.

Furthermore, the paper would benefit from a clear statement of its contribution to the existing literature, explicitly identifying any unresolved issues or suggesting future research directions.

Reviewer 1 ·

Basic reporting

- In the abstract, there is a statement without verb, "All critical, in safeguarding against cyber threats.", please check and revise.
- Also, in the abstract the author keep using "We" as in - "We delve.... ", "We categorize...."
- Using "We" is repeated more than 20 times over the text., which is not the best way to present, passive voice may be much better.
-Introduction needs rearrangement. In the current form, introduction is presented as one paragraph, which makes it difficult to get all the ideas presented. I would suggest to the authors to divide the introduction into a couple of paragraphs. Each paragraph should present a single idea (topic statement) followed by elaboration and explanation.
- The contents of the paper should definitely be presented in a separate paragraph.
- I am not sure about if it is a requirement of the journal to present the paper layout as in figure 1, if it is not, I would strongly recommend removing that figure. If it is a requirement, why the authors cite Wang et al as it appeared in the figure caption "Figure 1. Paper Outline Wang et al. (2023)."?

Experimental design

- Do the authors have to mention such details as the search string, the number of results, …. etc. If it is the journal requirement, it is ok.
- Are 20 papers sufficient for the purpose of the study?

Validity of the findings

- The paper in general reads well. However, in some discussions, the mentioned findings are either obvious or narrative.

·

Basic reporting

The manuscript is a well-structured and informative which address one of the most critical issues in modern cybersecurity. Clearly, the authors made good use of conciseness, making the material accessible to a wider audience. The introduction provides a solid foundation, clearly describing the problem and the proposed solution. The literature review is comprehensive, focuses on relevant research and demonstrates an in-depth understanding of the topic. The structure of the manuscript is consistent with academic standards, allowing the ideas to flow logically. The topic of network security, especially in software-defined networks, is of great interest to researchers, practitioners, and policymakers. This manuscript contributes to the ongoing discourse by providing new perspectives and valuable insights for further research.

Experimental design

Regarding the study design, the content of the manuscript is constant with the aim and scope of the journal. The authors achieved rigorous investigation of studies while adhering to excessive technical and ethical requirements. The literature review is comprehensive, properly-cited, and logically prepared into coherent paragraphs and subsections. The manuscript contributes to the sphere by means of inspecting a well-timed and applicable trouble, presenting treasured insight into the capability of software program-defined networks to enhance cybersecurity.

Validity of the findings

Once the findings are well explained, related to the motivations of the research, and based on supporting results, the paper would benefit from a thorough analysis of the impact and relevance of the findings If the contribution of the paper to the existing literature will be explicitly mentioned, which will enhance its relevance. In addition, identifying unresolved issues or future research directions will provide valuable directions for future studies in this area.

Additional comments

Here are some typos to improve manuscript quality:
• The introduction section is written in one long paragraph, it is better to divide it into two paragraphs and add a brief paragraph at the beginning of the introduction that gives the readers an idea about the importance of SDN in the field of networking. Also, I think there is no need to make a citation for the purpose of the paper in the fourth line of the introduction.
• It is preferable to change the title of Table3 to "Existing work in SDN Cybersecurity field "instead of "Existing work in this field".
• Although the language is clear, the grammar should be reviewed and verified.

Reviewer 3 ·

Basic reporting

Lack of Literature references.

Experimental design

Article content is within the Aims and Scope of the journal and article type.

Validity of the findings

Impact and novelty discussed

Additional comments

1) There are numerous review articles in this field. The necessity of the conducted research has not been justified in the introduction. What was the deficiency in the existing review articles that made it necessary to write this work?
2) (as shown in Figure 5. ----- >>>> (as shown in Figure 5).
3) In Table 3, it would be appropriate to include the names of the methods used in the existing works in the 'Solution' column. Add your own work in the row below and explain the advantages over the existing works.

---

## Round 0.2 · accepted · Accept

The authors have addressed the minor revisions requested by the reviewers and based on my review of the revised manuscript, I am recommending accepting this paper. It is not necessary to add the citations requested by the reviewer.

Reviewer 3 ·

Basic reporting

no comment

Experimental design

no comment

Validity of the findings

no comment

Additional comments

Add the following studies to the reference list:
1) Abdullayeva F.J., Suleymanzade S. Cyber security attack recognition on cloud computing networks based on Graph Convolutional Neural Network and GraphSage models, Results in Control and Optimization, 2024, vol. 15, pp. 1-10.
2) Abdullayeva F.J. Distributed denial of service attack detection in E-government cloud via data clustering, Array, 2022, vol. 15, pp. 1-12.